# Role of Continuous Glucose Monitoring in Supporting Glycemic Control Among Adolescents with Type 1 Diabetes in Saudi Arabia: A Retrospective Study

**DOI:** 10.3390/healthcare13050496

**Published:** 2025-02-25

**Authors:** Norah A. Alshehri, Maha Saud Alessa, Abdullah A. Alrasheed, Nada Alyousefi, Lemmese Alwatban, Haytham I. AlSaif, Ameerah Ali Alshehri

**Affiliations:** 1Department of Family and Community Medicine, College of Medicine, King Saud University (KSU), Riyadh 11362, Saudi Arabia; aalrasheed1@ksu.edu.sa (A.A.A.); nalyousefi@ksu.edu.sa (N.A.); lalwatban@ksu.edu.sa (L.A.); hayalsaif@ksu.edu.sa (H.I.A.); 2University Family Medicine Center, King Saud University Medical City, King Saud University, Riyadh 11495, Saudi Arabia; 3Family Medicine Department, King Saud University Medical City, Riyadh 12372, Saudi Arabia; maha.aliesa@gmail.com; 4Microbiology Department, Armed Forces Hospital, AlTaif 21944, Saudi Arabia; ameerahali2030@hotmail.com

**Keywords:** continuous glucose monitoring, HbA1c, type 1 diabetes, adolescents, glycemic control

## Abstract

**Background/Objectives:** Continuous glucose monitoring (CGM) has emerged as an important tool for the improvement of glycemic control in individuals with type 1 diabetes (T1D). However, its use has not been greatly explored among adolescents with special physiological and psychosocial challenges. This study evaluated the role of CGM in supporting glycemic control among high-risk adolescents with T1D in Saudi Arabia. **Methods:** This retrospective observational study was conducted among 73 adolescents aged 12–19 with T1D and baseline HbA1c ≥ 9% treated at King Khalid University Hospital in Riyadh, Saudi Arabia. Data were extracted from electronic health records over a three-month period. The HbA1c levels before and after CGM use were analyzed using paired *t*-tests. In addition, analyses included studying the correlations and regression models and assessing associations between changes in HbA1c and patient characteristics. **Results:** Mean HbA1c was significantly reduced from 9.48 ± 2.22% to 9.06 ± 1.91% following three months of CGM, with an average decrease of −0.42 ± 1.37%, *p* = 0.011. Changes in HbA1c did not correlate with various patient factors of interest: age, gender, body mass index, disease duration, and insulin type. Within a specified timeframe, 54.8% of patients reported hypoglycemia, and 38.4% reported diabetic ketoacidosis. **Conclusions**: CGM resulted in a small but statistically significant improvement in glycemic control in adolescents with T1D. Given such results, these findings highlight the need for larger, long-term trials to optimize CGM use in this vulnerable population, particularly through integrating advanced features (e.g., predictive alarms) and structured education programs to reduce hypoglycemia and DKA risks. Effective integration of CGM in daily diabetes management may lead to better long-term clinical outcomes and improved quality of life for adolescents.

## 1. Introduction

Type 1 diabetes (T1D) is a chronic autoimmune disorder characterized by the destruction of insulin-producing β-cells in the pancreas, resulting in absolute insulin deficiency [1]. It affects approximately 9.5 million individuals worldwide, with an increasing incidence, particularly among adolescents [2]. The burden of T1D is significant, requiring lifelong insulin therapy and continuous blood glucose monitoring to maintain optimal glycemic control. Suboptimal glycemic control is the major reason for diabetes complications such as neuropathy, nephropathy, retinopathy, and cardiovascular disease [3]. Maintaining HbA1c levels below 7.0%, as recommended by the American Diabetes Association (ADA), will delay or prevent long-term complications [4]. However, it is particularly challenging in adolescents because of physiological insulin resistance in puberty, variable adherence to insulin therapy, psychosocial stressors, and lifestyle variability [5]. Recent studies show that adolescents with T1D face greater glycemic fluctuations than adults, leading to increased long-term cardiovascular risk and a higher likelihood of diabetic ketoacidosis (DKA) episodes [6].

Since the introduction of self-monitoring of blood glucose (SMBG), traditional (manual) glucose measurements have remained the gold standard in glycemic management. Finger-prick samples are required for SMBG, and while glucose is measured at only a one-time point, this can offer some value over a short duration; however, it cannot allow one to deduce continuous trends of glucose or touch on predictive information [6]. The continuous glucose monitoring (CGM) system has revolutionized the management of diabetes by providing real-time continuous glucose monitoring. CGM devices, such as the FreeStyle Libre, measure interstitial glucose levels every 5–15 min. They enable patients and healthcare providers to monitor glucose excursions, receive alerts for hypoglycemia, and analyze time-in-range (TIR) data [7]. Whereas SMBG only provides retrospective information, CGM makes it possible to make proactive changes to avoid excursions in glucose levels.

Real-world data demonstrate the beneficial effect of CGM on glycemic control improvement. Trial data show that CGM use is associated with a reduction in HbA1c of 0.5–1.1%, glucose variability, and incidence of hypoglycemia and hyperglycemia compared to SMBG [8]. CGM is even more valuable when paired with insulin pumps, allowing insulin delivery, which improves time in the target glucose range [9]. While adults with T1D have responded strongly to improved HbA1c levels through the use of CGM, trials have had variable effectiveness in adolescents, potentially because of inconsistency in adherence, sensor wear time, and behavior [10]. A recent real-world study found that only 40% of adolescents consistently wear CGM sensors, limiting its potential benefits [11]. Current CGM users report lower levels of diabetes distress compared to past users, indicating a potential benefit of CGM in reducing the emotional burden associated with diabetes management [12].

The prevalence of T1D in Saudi Arabia is one of the highest in the world, at an estimated 35.4 per 100,000 children per year [13]. Despite the growing use of CGM in Saudi Arabia, there remains limited real-world data on the effectiveness of CGM in adolescents [14,15]. Fewer than four studies have specifically evaluated CGM use in Saudi adolescents, and even fewer have analyzed predictors of glycemic improvement such as gender, BMI, disease duration, and insulin regimen. Given the unique socio-cultural factors affecting diabetes management in Saudi Arabia, there is a critical need to assess CGM outcomes in this underrepresented population. The assessment of CGM’s effect on glycemic control in Saudi T1D adolescents is important to inform clinical practice, enhance diabetes education, and ultimately optimize long-term outcomes.

The primary objective of this study is to assess the role of CGM in supporting glycemic control by comparing HbA1c values before and after the use of CGM in adolescents with T1D in Saudi Arabia.

The secondary objective is to investigate the predictors of change in HbA1c following CGM initiation, with a particular focus on the role of gender, BMI, disease duration, and insulin regimen to provide insights into personalized diabetes management.

## 2. Materials and Methods

### 2.1. Aims and Research Questions

The overall aim of this study is to evaluate the impact of continuous glucose monitoring (CGM) on glycemic control and identify factors influencing its effectiveness among adolescents with type 1 diabetes (T1D) in Saudi Arabia.

Primary Research Question:

Does the use of CGM improve glycemic control, as measured by the change in HbA1c values before and after CGM initiation in adolescents with T1D?

Secondary Research Question:

What are the predictors of changes in HbA1c following the initiation of CGM?

### 2.2. Study Design, Participants, and Setting

This retrospective pre–post observational study was conducted between 1 January and 31 March 2024 at the diabetes clinics of King Khalid University Hospital (KKUH) in Riyadh, Saudi Arabia.

The study population included eligible participants who met the following inclusion criteria:Adolescents aged 12–19 with T1D;A diagnosis of T1D according to the American Diabetes Association (ADA) criteria, including clinical presentation, fasting hyperglycemia (>126 mg/dL), presence of diabetes-related autoantibodies (e.g., GAD-65, IA-2, ZnT8), and low/undetectable C-peptide levels (4);HbA1c levels ≥ 9%;Patients who exclusively used the conventional finger-pricking method for glucose monitoring before initiating CGM with the FreeStyle Libre system for the first time. CGM was used for at least 12 weeks before the start of the study period. CGM was conducted using the FreeStyle Libre system (Abbott Diabetes Care), a factory-calibrated, intermittently scanned CGM device that records interstitial glucose levels every 15 min without requiring routine finger-prick calibration. The system consists of a small sensor (14-day lifespan) applied to the back of the upper arm, which continuously measures glucose concentrations in interstitial fluid. Data can be accessed by scanning the sensor with a handheld reader or smartphone application.

Patients with severe diabetes complications, significant medical or psychiatric comorbidities, or pregnancy were excluded to minimize confounding effects and ensure this study’s focus on glycemic outcomes.

### 2.3. Sampling Method

In this study, a non-probability consecutive sampling method was adopted for to recruit patients. All patients who presented within the specified period and fulfilled the inclusion criteria were included in this study. A total of 73 patients were studied. The reason for selecting this method is that the target population is relatively small, which means that this method could ensure that the sample represented all eligible patients. In the selection process, all subjects fulfilling the criteria were included to ensure minimal selection bias; thus, this also improves the representativeness, maximizing the generalization of the findings within the defined population.

### 2.4. Data Collection Procedure

Data were retrospectively taken from electronic medical records in the KKUH E-SIHI. All available data concerning visits to a diabetes clinic were manually drawn and recorded by the researcher on a data collection sheet prepared beforehand. For this research, we began to gather data upon receiving approval from the IT department and the Institutional Review Board (IRB), No. E-23-8291.

### 2.5. Instrument Tool

A structured data sheet developed for this study included all the information that needed to be collected. Demographic and clinical variables were noted down according to the objectives of this study. These variables included patient serial number, age, gender, BMI, HbA1c levels (before and after the use of CGM for 12 weeks), hypoglycemia and DKA occurrence, disease duration, and different insulin regimens, including multiple daily injections (MDIs) and continuous subcutaneous insulin infusion (CSII) via insulin pumps. The type of insulin therapy was documented for each participant to minimize bias (e.g., long-acting insulin and short-acting insulin). BMI categories were classified according to the WHO standard intervals (underweight, normal, overweight, and obese) [16]. The insulin protocol used in this study was a basal-bolus therapy, with the insulin types categorized as either long-acting or short-acting. Long-acting insulins (Lantus, Tresiba, and Toujeo) provided basal coverage and were administered once daily to maintain baseline glucose levels. Short-acting insulins (Aspart and Humalog) were used for bolus dosing before meals or during glucose spikes. The insulin regimen was tailored to individual patient needs, with doses adjusted based on clinical parameters and changes in blood glucose levels.

A data sheet was developed based on an extensive literature review, with expert consultation from active endocrinologists and biostatisticians to provide a comprehensive list of necessary variables. The design aimed to maximize usability, accuracy, and relevance for this study.

The content validity of the data sheet was assessed by a panel of experts in family medicine and diabetes management. Each field was reviewed to ensure it aligned with this study’s objectives and captured reliable and clinically meaningful data. Pilot testing was conducted on a small sample of patients to evaluate the clarity, feasibility, and efficiency of data collection. Feedback from the pilot test was incorporated to refine the tool, ensuring its reliability and minimizing potential data entry errors.

This study follows the Strengthening the Reporting of Observational studies in Epidemiology (STROBE) guidelines [17] to ensure transparency and reproducibility [DOI: 10.1016/j.jclinepi.2007.11.008]. A completed STROBE checklist is provided in the Appendix A.

### 2.6. Ethical Considerations

This study was conducted in accordance with the ethical principles of the Declaration of Helsinki and was approved by the Institutional Review Board in advance (IRB Approval No. E-238291). The data were anonymized using numerical coding to maintain patient confidentiality, and all data collected were used only for this research.

### 2.7. Statistical Data Analysis

Analysis was performed with IBM SPSS Statistics, Version 28. Numerical data are shown as the mean ± SD, while categorical data are expressed as frequencies and percentages.

Comparisons of HbA1c levels between pre- and post-CGM implementation were made using paired *t*-tests. Comparisons between two groups were performed using unpaired *t*-tests, while differences among more than two groups were analyzed using one-way ANOVA. Pearson’s correlation coefficient was calculated to assess the relationships between continuous variables.

A linear regression analysis was conducted to assess the factors associated with changes in HbA1c after the implementation of CGM, adjusting for variables such as age, baseline HbA1c, BMI, and treatment regimen. The two-tailed *p*-value was considered statistically significant if <0.05.

## 3. Results

### 3.1. Baseline Characteristics of Adolescents with Type 1 Diabetes

The study sample included 73 adolescents with T1D, aged 12–20 years (mean age: 14.62 ± 2.28 years). The gender distribution was nearly equal, with 50.7% male and 49.3% female participants. The mean duration of diabetes was 5.31 ± 3.59 years. Regarding BMI, 42.5% of the participants had normal BMI values, while 20.5% were underweight. Among long-acting insulin therapies, Lantus was the most frequently prescribed (72.6%), followed by Tresiba (13.7%) and Toujeo (11.0%). Additionally, nearly all participants (97.3%) received Aspart as their short-acting insulin (Table 1).

### 3.2. Impact of Continuous Glucose Monitoring in Supporting Glycemic Control

HbA1c levels showed a statistically significant reduction after 12 weeks of CGM implementation (*p* = 0.011). The mean HbA1c decreased from 9.48 ± 2.22% before CGM to 9.06 ± 1.91% after CGM, with an average change of −0.42 ± 1.37% (Table 2).

Following CGM implementation, 54.8% of the participants experienced hypoglycemia, while 45.2% reported no hypoglycemic events. Diabetic ketoacidosis (DKA) was reported in 38.4% of the participants, whereas 57.5% had no DKA episodes.

### 3.3. Factors Associated with Changes in HbA1c After CGM Implementation

Table 3 shows the relationship between gender, BMI, and insulin use with HbA1c levels before and after CGM. While reductions in HbA1c were observed across all groups, these changes were not statistically significant (e.g., *p* = 0.321 for gender, *p* = 0.704 for BMI, and *p* = 0.792 for long-acting insulin). The largest reduction was seen in the underweight BMI group (−0.90 ± 1.81%) and among Toujeo users (−0.61 ± 0.86%).

No significant correlations were observed between HbA1c changes and age (r = 0.004, *p* = 0.973) or diabetes duration (r = 0.016, *p* = 0.895), as shown in Table 4.

No significant relationships were observed in the univariate and multivariable regression analyses regarding changes in HbA1c following CGM implementation based on gender, BMI categories, or type of insulin (Table 5). For example, gender showed no significant effect (coefficient: 0.13; 95% CI: −0.66 to 0.93; *p* = 0.739), and nor did BMI (coefficient: 0.64; 95% CI: −0.36 to 1.62; *p* = 0.205) (Table 5).

## 4. Discussion

This study evaluated the role of continuous glucose monitoring (CGM) in supporting glycemic control in adolescents with type 1 diabetes (T1D) in Saudi Arabia. The results showed a statistically significant reduction in HbA1c levels following 12 weeks of CGM implementation. The total HbA1c level reduced from 9.48% to 9.06%, with the mean reduction being −0.42% (*p* = 0.011). Although the extent of the reduction varied, these findings document the potential of CGM to exert a moderate effect on glycemic control over a short interval.

Hypoglycemia was experienced by 54.8% of participants, emphasizing the continued difficulty in reconciling glycemic control and the prevention of adverse events. Diabetic ketoacidosis (DKA) was experienced by 38.4% of participants after starting CGM, emphasizing the necessity for multifaceted management strategies in conjunction with CGM use.

Subsequent analysis revealed that the largest drops in HbA1c occurred among the underweight (−0.90%) and the patients on Toujeo (−0.61%), although these were not statistically significant. No correlations existed between demographic or clinical variables (i.e., gender, BMI, or insulin regimen) and changes in HbA1c. Regression analysis established that the impact of CGM on HbA1c was unaffected by these variables.

The significant reduction in HbA1c levels (−0.42%, *p* = 0.011) in this study is in line with previously published studies on the benefits of continuous glucose monitoring (CGM) in supporting glycemic control in adolescents with type 1 diabetes. HbA1c reduction is independent of the mode of insulin delivery used, as proven by Teoh et al., thus highlighting the adaptability of CGM in different clinical contexts [18]. In addition, Laffel et al. have described a substantial but small decrease in HbA1c after the use of CGM for 26 weeks, which is of the same magnitude as the change observed in our short-term trial [19].

In addition, a systematic review conducted by Teo et al. corroborated the advantages of continuous glucose monitoring (CGM) compared to self-monitoring of blood glucose (SMBG) in enhancing glycemic control, especially in patients with higher baseline HbA1c levels, similar to our study population’s high baseline levels [20].

However, the decrease in HbA1c demonstrated here is less profound than that from long-term trials. Dorando et al. documented larger decreases in HbA1c in longer follow-up studies, suggesting that CGM effects would build with chronic use and adaptive behavior [21]. In addition, Huhn et al. proposed that the magnitude of the decrease in HbA1c could be influenced by patients’ interpretation and response to CGM data [11]. The efficacy of CGM in enhancing glycemic control lies in real-time feedback, allowing the patient to recognize glucose swings and adjust insulin promptly. Unlike SMBG, which provides point-check data, CGM supplies constant data, improving pattern recognition and making it easier for adolescents to manage diabetes.

The very high rate of hypoglycemia (54.8%) after the initiation of CGM is concerning since the prevention of hypoglycemic episodes is one of the most significant goals of CGM. Messaaoui et al. demonstrated that CGM was much more protective against severe hypoglycemia compared to flash glucose monitoring [22]. In the same vein, Marigliano et al. demonstrated that alarm notification-based CGM systems can substantially reduce the risk of hypoglycemia [23]. This highlights the necessity of combining CGM with other technological aspects or formal education to maximize its potential to avoid hypoglycemia [24].

Within the setting of diabetic ketoacidosis (DKA), our rate of 38.4% is higher than rates shown by other CGM studies. Dorando et al. illustrated that longer CGM use and frequent data interpretation can greatly reduce DKA events [21]. Choudhari et al. pointed out that the real-time and remote application of CGM in adolescents with poorly controlled diabetes could reduce overall glycemic instability and complications such as DKA [25]. In contrast, our higher DKA rates may be the result of a short follow-up period.

Furthermore, Borbjerg et al. highlighted the critical role of timely CGM initiation, suggesting that delays in adopting CGM can result in less optimal glycemic control, similar to our study population’s mean duration of T1D of 5.31 years [26]. This indicates that earlier implementation of CGM, coupled with proper patient education and feedback mechanisms, is essential to minimize the risk of both hypoglycemia and DKA, as also demonstrated by Marigliano et al. [23].

Overall, the findings from this study add to the mounting evidence in favor of CGM as an effective tool for enhancing glycemic control. Nevertheless, as Williams et al. and Messer et al. have implied, CGM’s value lies not only in clinical benefit but also in advantages such as better self-management, psychological well-being, and satisfaction with treatment [27,28]. The inclusion of behavioral interventions and bridging the technological gap can increase its value to diabetes management.

### 4.1. Perspectives for Clinical Practice

A key strength of this study is its focus on adolescents with type 1 diabetes, a population known for challenges in achieving optimal glycemic control due to developmental and behavioral factors. This study also provides real-world evidence of CGM’s impact over a 12-week period, highlighting practical implications for clinical practice.

The results of this research endorse the utilization of CGM in routine care in type 1 diabetes adolescents to realize improved glycemic control. However, maximum benefits from CGM can be derived only through a multi-dimensional and multi-disciplinary intervention, especially in schools, where the majority of adolescents’ time is spent.

One of the most critical elements of diabetes care in the school setting is the school nurses’ role in managing children and adolescents with T1D. School nurses are the primary facilitators of diabetes care plan adherence.

The integration of CGM technology into school health programs can significantly enhance diabetes management by enabling real-time glucose level monitoring, timely intervention, and ready communication between healthcare providers, parents, and students.

School personnel and nurses must undergo a well-structured training program to achieve optimal utilization of CGM for diabetes management in schools. This includes training in CGM interpretation, adjustment of insulin dosing, emergency protocol, and communication with the family and healthcare team. A standardized school-based diabetes care policy can also optimize care consistency and outcomes in adolescents.

Nevertheless, to derive maximal benefit, intensive educational sessions on the correct interpretation of data and appropriate adjustment of insulin doses must be provided [29]. The removal of complications such as hypoglycemia through the addition of new CGM features such as predictive alarms can also improve patient safety and treatment outcomes [30,31]. The long-term effect of CGM on glycemic variability, hypoglycemia, and DKA would be the ideal subject of future studies on large and diverse populations. The combination of CGM with other novel technologies, such as automated insulin delivery systems, and complementing it with behavioral interventions and family support programs is likely to be the secret to maximizing long-term outcomes and quality of diabetes care in general.

### 4.2. Limitations

However, several limitations should be acknowledged. As suggested by previous studies, the relatively short follow-up period may have limited the full potential of CGM to manifest long-term glycemic improvements. Additionally, the small sample size and the single-center design may limit the generalizability of the results to broader populations. Other autoimmune diseases, such as thyroid disorders or celiac disease, were not considered or assessed in this study.

## 5. Conclusions

This study found that continuous glucose monitoring can help support the improvement in glycemic control in T1D adolescents through a reduction in HbA1c of a modest 0.42% in three months. Although limited by a small sample size and retrospective nature, these data support the use of CGM as part of regular care in high-risk adolescents. Larger studies with longer controlled trials are necessary to establish longer-term benefits as well as raise concerns regarding adherence. CGM represents an emerging therapy for optimizing outcomes and minimizing complications in adolescents with T1D.

## Figures and Tables

**Table 1 healthcare-13-00496-t001:** Baseline characteristics of adolescents with type 1 diabetes in this study.

Item	N	%
Age (years), mean ± SD	14.62 ± 2.28 (12–20)	
Gender		
Male	37	50.7
Female	36	49.3
BMI (kg/m^2^)		
Underweight	15	20.5
Normal	31	42.5
Overweight	12	16.4
Obesity 1	10	13.7
Obesity 3	2	2.7
N/A	3	4.1
Disease duration (years)		
Mean ± SD	5.31 ± 3.59 (0.33–13)	
Long-acting insulin		
Lantus	53	72.6
Tresiba	10	13.7
Toujeo	8	11.0
N/A	2	2.7
Short-acting insulin		
Aspart	71	97.3
Humalog	1	1.4
N/A	1	1.4

BMI: body mass index.

**Table 2 healthcare-13-00496-t002:** Changes in HbA1c levels before and after continuous glucose monitoring (CGM).

Item	HbA1c (%)		Change	*p*-Value
	Before CGM	After CGM		
Mean ± SD	9.48 ± 2.22	9.06 ± 1.91	−0.42 ± 1.37	0.011
Range	5.40–14.80	5.50–14	−6.30 to 2.90	

Statistical significance at *p*-value < 0.05, CGM: Continues glucose monitoring, HbA1c: Glycated hemoglobin.

**Table 3 healthcare-13-00496-t003:** Relationship between (gender, BMI, and medication use) of patients and their HbA1c levels.

Item	HbA1c (%)		Change	*p*-Value (Change)
	Before CGM	After CGM		
Gender				
Male	9.82 ± 2.45	9.25 ± 2.01	−0.58 ± 1.41	0.321
Female	9.13 ± 1.92	8.87 ± 1.81	−0.26 ± 1.32	
BMI (kg/m^2^)				
Underweight	10.17 ± 2.1	9.27 ± 1.6	−0.9 ± 1.81	0.704
Normal	9.34 ± 2.45	9.06 ± 2.27	−0.27 ± 1.42	
Overweight	9.29 ± 1.6	8.93 ± 1.31	−0.37 ± 0.57	
Obesity 1	8.57 ± 1.94	8.26 ± 1.11	−0.31 ± 1.51	
Obesity 3	10.4 ± 0	10.2 ± 0	−0.2 ± 0	
Long-acting insulin				
Lantus	9.64 ± 2.26	9.18 ± 1.94	−0.45 ± 1.48	0.792
Tresiba	8.58 ± 1.53	8.4 ± 1.74	−0.18 ± 1.29	
Toujeo	10.19 ± 2.4	9.58 ± 1.85	−0.61 ± 0.86	
Short-acting insulin				
Aspart	9.55 ± 2.21	9.11 ± 1.91	−0.43 ± 1.39	---
Humalog	8.4	8.5	0.1	

Data are presented as mean ± SD; statistical significance at *p*-value < 0.05.

**Table 4 healthcare-13-00496-t004:** Correlation between (age and DM duration) of patients and their HbA1c levels.

Item		HbA1c (%)		Change
		Before CGM	After CGM	
Age (years)	r	−0.011	−0.010	0.004
	*p*-value	0.926	0.933	0.973
Duration of DM (years)	r	0.105	0.132	0.016
	*p*-value	0.381	0.271	0.895

r: Pearson’s correlation coefficient; statistical significance at *p*-value < 0.05.

**Table 5 healthcare-13-00496-t005:** Linear regression analysis for factors associated with the change in HbA1c levels after CGM.

Item	Univariate Analysis			Multivariable Analysis		
	Coefficient	95% CI	*p*-Value	Coefficient	95% CI	*p*-Value
Age (years)	0.002	−0.14 to 0.14	0.973	−0.02	−0.19 to 0.14	0.785
Gender						
Male	Ref			Ref		
Female	0.32	−0.32 to 0.96	0.321	0.13	−0.66 to 0.93	0.739
BMI (kg/m^2^)						
Underweight	Ref			Ref		
Normal	0.63	−0.26 to 1.52	0.165	0.64	−0.36 to 1.62	0.205
Overweight	0.53	−0.56 to 1.63	0.335	0.49	−0.85 to 1.82	0.467
Obesity 1	0.59	−0.56 to 1.74	0.311	0.55	−0.77 to 1.87	0.409
Obesity 3	0.7	−1.43 to 2.83	0.514	0.7	−1.48 to 2.89	0.522
Disease duration (years)	0.01	−0.08 to 0.1	0.895	0.003	−0.1 to 0.11	0.952

CI: confidence interval; statistical significance at *p*-value < 0.05.

## Data Availability

Data used in this study are available upon reasonable request from the corresponding author due to privacy/ethical restrictions.

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
