# Peer review of "Role of Continuous Glucose Monitoring in Supporting Glycemic Control Among Adolescents with Type 1 Diabetes in Saudi Arabia: A Retrospective Study"

_healthcare, 2025, doi:10.3390/healthcare13050496_

Round 1
Reviewer 1 Report (Previous Reviewer 1)
Comments and Suggestions for Authors
None.
Author Response
We sincerely thank you for your thorough review of our manuscript and for taking the time to assess our revisions. We are pleased to hear that you found the changes satisfactory and have no further comments.
Your feedback has been invaluable in helping us improve the quality of our work.
Reviewer 2 Report (Previous Reviewer 2)
Comments and Suggestions for Authors
The authors made several changes in their manuscript.
The current version is much better than the first one.
Here are some other revisions in order to improve it before publication :
- use abreviations in abstract
- in titles : no need to specify T1D adolescents
- delete the definition of BMI intervals in the results and tables
- Table 1 could be summerized in fewer lines
- table 3 not useful
- table 4 : define insulin protocols ; how were they used
- more abreviations could be used
minor editing
Author Response
Please see the attachment.

Reviewer 3 Report (Previous Reviewer 3)
Comments and Suggestions for Authors
Dear Authors,
the comments in the annex file.
Best

I suggest native review
Round 2
Reviewer 3 Report (Previous Reviewer 3)
Comments and Suggestions for Authors
Dear Authors,
in this form the manuscrit is certly almost ready for publication. I suggets only in the new part of discussion 4.1 Perspectice for clinical practice to expand the discussion with multi-dimensional and multi-disciplinary management view of T1D in particular setting as school. I suggest to extend the discussion in theme as "Impact of School Nures of managing Pediatric Type 1 Diabetes with technological devices support".
Author Response
Please see the attachment.

This manuscript is a resubmission of an earlier submission. The following is a list of the peer review reports and author responses from that submission.
Round 1
Reviewer 1 Report
Comments and Suggestions for Authors
This is a retrospective study with a small sample size. This study investigated the effects of CGM in improving glycaemic control in adolescents with T1DM. This is an interesting topic. However, there are some problems with this article. The method of using CGM should be explained in detail. The discussion part is too short. The author should provide more analysis and comments. The data is indeed too limited, the author should add other relevant data to support the results.
Comments on the Quality of English LanguageThe English could be improved to more clearly express the research.
Reviewer 2 Report
Comments and Suggestions for Authors
The authors analyzed in a retrospective study the efficiency of CGM in improving diabetes in adolescent patients.
The study lacks several important aspects to be accepted in this form. I strongly advise the authors to improve and rewrite somes parts and resubmit it.
Here are my comments :
- CGM is a measure tool, that could not be efficient on glycemic control, but help monitoring glycemia by using pumps or insulin analogues. The title must be changed to highlight this aspect.
- Introduction is too short, and we could not highlight the difference between CGM and other measurement tools in diabetes.
- Methods must raise the definition of different insluin protocols that could highly biase the diabetes control : pumps are more efficient than other tools.
- Diabetes duration also is important with other diabetes comorbidities that could interfere with glycemic control.
- Diagnosis of type 1 diabetes made how.
- Other autoimmune diseases ?
- The results are not pertinent in most of them. Analyzing diabetes control with other associated factors such as Height or LDL is not important.
The authors analyzed several aspects that could not add any pertinent conclusions on how CGM is important in these patients.
- Tables are numerous, could be fused.
- Explain the use of Metformin in type 1 diabetes ?
- Explain also how some patients were obese and how autoimmune diabetes was suspected in them.
- Discussion is too short and must focus on the sepcific role of CGM compared to other tools.
- References are limited : only 10. Many other references are published recently on the subject and could be used to enhance discussion.
Comments on the Quality of English LanguageMinor editing
Reviewer 3 Report
Comments and Suggestions for Authors
Dear Authors,
the comments in the annex file.
Best

The native english review suggest